# A new perspective and assessment measure for common dissociative experiences: 'Felt Sense of Anomaly'

Emma Černis[1]*, Esther Beierl[2], Andrew Molodynski[3], Anke Ehlers[2,3‡],
Daniel Freeman[1,3‡]

**1** Department of Psychiatry, University of Oxford, Oxford, United Kingdom, **2** Department of Experimental Psychology, University of Oxford, Oxford, United Kingdom, **3** Oxford Health NHS Foundation Trust, Oxford, United Kingdom

‡ These authors are joint senior authors on this work.
* emma.cernis@psych.ox.ac.uk

**Citation:** Černis E, Beierl E, Molodynski A, Ehlers A, Freeman D (2021) A new perspective and assessment measure for common dissociative experiences: 'Felt Sense of Anomaly'. PLoS ONE 16(2): e0247037. https://doi.org/10.1371/journal.pone.0247037

**Data Availability Statement:** Data cannot be shared publicly because of the terms and conditions contained within the ethics permissions granted for this study from the Central Research

## Abstract

### Background

Dissociative experiences occur across a range of mental health disorders. However, the term 'dissociation' has long been argued to lack conceptual clarity and may describe several distinct phenomena. We therefore aimed to conceptualise and empirically establish a discrete subset of dissociative experiences and develop a corresponding assessment measure.

### Methods

First, a systematic review of existing measures was carried out to identify themes across dissociative experiences. A theme of 'Felt Sense of Anomaly' (FSA) emerged. Second, assessment items were generated based on this construct and a measure developed using exploratory (EFA) and confirmatory (CFA) factor analyses of 8861 responses to an online self-report survey. Finally, the resulting measure was validated via CFA with data from 1031 patients with psychosis.

### Results

'Felt sense of anomaly' (FSA) was identified as common to many dissociative experiences, affecting several domains (e.g. body) and taking different forms ('types'; e.g. unfamiliarity). Items for a novel measure were therefore systematically generated using a conceptual framework whereby each item represented a type-by-domain interaction (e.g. 'my body feels unfamiliar'). Factor analysis of online responses found that FSA-dissociation manifested in seven ways: anomalous experiences of the self, body, and emotion, and altered senses of familiarity, connection, agency, and reality ($X^2$ (553) = 4989.435, p<0.001, CFI = 0.929, TLI = 0.924, RMSEA = 0.052, SRMR = 0.047). Additionally, a single-factor 'global FSA' scale was produced ($X^2$ (9) = 312.350, p<0.001, CFI = 0.970, TLI = 0.950, RMSEA = 0.107, SRMR = 0.021). Model fit was adequate in the clinical (psychosis) group ($X^2$ (553) =

Ethics Committee of the University of Oxford, the NHS Research Ethics Committee, and Health Research Authority, and consented to by participants. Surveys were confidential to enable freedom of expression by participants, and participants consented into the study without being consulted as to the sharing of anonymised data. Therefore, only descriptive statistics, which qualify as the minimal data set, are included in the paper.

**Funding:** This study was funded by a Wellcome Trust Clinical Doctoral Fellowship awarded to EČ (102176/B/13/Z https://wellcome.ac.uk). AE is funded by the Wellcome Trust (200796 https://wellcome.ac.uk), the Oxford Health NIHR Biomedical Research Centre (BRC-1215-20005) and an NIHR Senior Fellowship. DF is funded by an NIHR Research Professorship (RP-2014-05-003 https://www.nihr.ac.uk). The views expressed are those of the authors and not necessarily those of the NHS, the NIHR or the Department of Health. The funders had no role in study design, data collection and analysis, decision to publish, or preparation of the manuscript.

**Competing interests:** he authors have declared that no competing interests exist.

1623.641, p<0.001, CFI = 0.927, TLI = 0.921, RMSEA = 0.043, SRMR = 0.043). The scale had good convergent validity with a widely used dissociation scale (DES-II) (non-clinical: r = 0.802), excellent internal reliability (non-clinical: Cronbach's alpha = 0.98; clinical: Cronbach's alpha = 0.97), and excellent test-retest reliability (non-clinical: ICC = 0.92). Further, in non-clinical respondents scoring highly on a PTSD measure, CFA confirmed adequate model fit ($X^2$ (553) = 4758.673, CFI = 0.913, TLI = 0.906, RMSEA = 0.052, SRMR = 0.054).

## Conclusions

The Černis Felt Sense of Anomaly (ČEFSA) scale is a novel measure of a subset of dissociative experiences that share a core feature of FSA. It is psychometrically robust in both non-clinical and psychosis groups.

## Introduction

*'Some have criticized the concept of dissociation itself, pointing out that it has become over-inclusive and therefore meaningless [. . .] Between critics and specialists yawns an unbridged chasm, so that the field has remained in disconnected state'* [1].

Since Janet's influential work [2], which outlined dissociation as an altered state of consciousness resulting from traumatic events, the array of phenomena encompassed within the term dissociation has expanded to such an extent that–as the quotation above highlights–any unifying concept has become obscured. This lack of clarity, combined with the often difficult to describe nature of the phenomena, makes dissociation a challenging field of mental health research. Because dissociation has become 'a vague term used to describe a broad range of phenomena' [3], theorists, clinicians, and researchers may be using the same term to refer to rather different phenomena, depending on which–often unstated–assumptions are being made. This contributes to the continued under-recognition and misidentification of dissociation clinically [4], and impedes progress in research [5,6]. Therefore, this paper seeks to define a circumscribed area within the broad concept of "dissociation", delineate precisely which phenomena fall within this category, and develop a corresponding measure to facilitate its study.

In response to the heterogeneity, several theorists have taken the approach of suggesting that sub-categories of dissociative experience exist. Most notably, Holmes et al. [7] propose that there are two distinct forms of dissociation: detachment and compartmentalisation. The former describes experiences involving altered states of consciousness, such as depersonalization, derealisation and other forms of separation from one's internal or external environment. The latter–compartmentalization–refers to deficits or loss of control in specific functions, such as in dissociative amnesia, dissociative seizures, or functional neurological symptoms. Holmes and colleagues [7] state that although both forms of dissociation may exist on a spectrum of severity, they are nevertheless independent and need not co-occur. By referring to both forms as 'dissociation', we may be conflating two separable phenomena.

In this study, we therefore propose to seek out phenomenological subcategories of dissociative experience *de novo*, using multiple sources of information, and without prior hypotheses as to what distinctions may arise. This approach follows that taken by clinician-researchers such as Clark and Ehlers [8,9], whose translational treatment-development work demonstrates that before a theoretical basis for understanding a particular phenomenon can exist, it must first be clearly understood at the phenomenological level.

At present, the majority of research uses the Dissociative Experiences Scale as a measure of dissociation (DES; [10,11]). This is the longest-standing and most widely-used measure of dissociative experiences, containing 28 items such as 'Some people are told that they sometimes do not recognize friends or family members' and 'Some people find that they sometimes are able to ignore pain'. Whilst this measure has had significant impact in the field and greatly facilitated discourse about dissociative experiences in clinic and research, the DES does have limitations [7,12]. Most relevant here is the observation that the DES omits some experiences (most notably emotional numbing) that would be required for a comprehensive measurement of dissociation. Accordingly, it would be beneficial to research and clinical endeavours if any new characterisations of dissociative sub-categories were accompanied by a comprehensive measure of that construct.

Therefore, we describe here a novel definition of a category of dissociative experiences using a patient-informed, data-driven approach, and then develop its corresponding measure.

## A systematic review of phenomenology

In the absence of a consensus regarding the symptoms and mechanisms of dissociation, we first sought to identify a coherent set of experiences on the basis of the phenomenology studied to date under the term. To achieve this, a systematic search of the literature for measures of dissociation was undertaken (See Table 1 for search terms and Fig 1 for the PRISMA diagram; the search and data extraction was performed by EČ). Measures were chosen since these must necessarily specify which phenomena are most relevant or prototypical when assessing the concept to be measured, and therefore should provide descriptions of notable, fundamental examples of dissociative phenomenology. Specifically, papers were sought where a measure of dissociation (or an incorporated concept, e.g. depersonalization) was subjected to factor analysis. The aim was to inspect the factors produced by these analyses and search for common themes among measures.

Table 2 summarises the 77 papers which factor analysed 26 measures of dissociation. The DES received the most attention of any individual measure, with 28 factor analyses carried out on the adult version of this scale. Of these, just over half found absorption (n = 19) and depersonalization (n = 18) were a factor in dissociation; half incorporated some form of memory difficulty or amnesia; and seven found a single factor structure. By contrast, non-DES measures (41 studies, 24 measures) were more mixed, and less likely to incorporate absorption (4 studies, 2 measures), or memory problems (11 studies, 9 measures). However, factors relating to depersonalisation experiences were still relatively common (present in 20 studies of 13

**Table 1. Summarising the search method and results of the systematic review of existing dissociation measure studies.**

| *Method* |
|---|
| The search was run on 15th July 2019 in Ovid Medline using the search terms: dissociation; dissociative; depersonali*; dereali*; "intrusi* + memor*"; flashback*; unreality; fugue; reliving; "conversion+disorder". (Note that the wildcard "dissociat*" was not used as this returns many papers entitled "Dissociating. . . [X] and [Y]. . ." which are not in the dissociation or wider clinical psychology literature). |
| The search was limited to English language, and to journal article or review formats. Due to the occurrence of the word 'dissociation' used in other contexts (as above), search terms were limited to the title and abstract of the paper, ensuring that a full-text search did not pick up irrelevant uses of the terms. |
| *Results* |
| Despite the conservative approach to search criteria, a large number of irrelevant results were produced. Therefore, a total of 14474 papers were retrieved meeting the above criteria. Titles and abstracts were then searched by hand using Mendeley Reference Manager (v.1.19.2) to identify relevant papers. This produced a smaller group of 138 papers discussing the measurement of dissociation (Fig 1). |

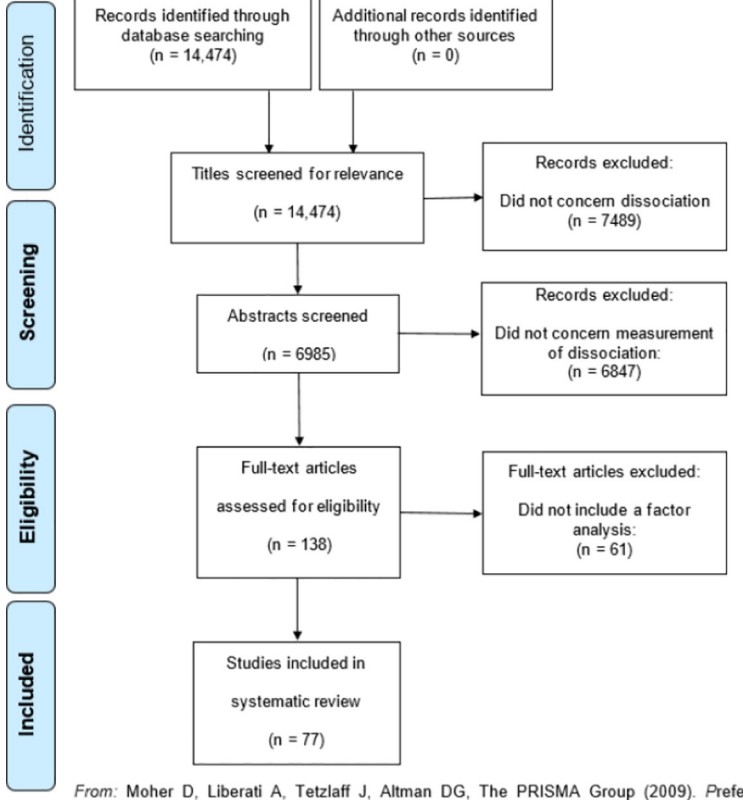

**Fig 1. PRISMA 2009 flow diagram for the systematic review of dissociation measure studies.**

measures), and a similar proportion of factor analyses resulted in a single factor structure (12 studies of 6 measures). Across Table 2, excluding single factor results, approximately 70 unique factors have been implicated in dissociation.

Table 2 illustrates the argument that experiences described as "dissociative" cover such a wide range of domains and processes that these are now difficult to unify completely in an understandable way. Although experiences of derealization, depersonalisation and amnesia were described by a number of measures, Table 2 shows no unanimous inter-measure themes of phenomenology.

## Definition & framework development

In order to identify a common denominator for a proportion of people's dissociative experiences, the dissociation measures identified in the systematic review above were examined. This inspection found that many items of these measures contain words which imply the presence of a 'felt sense of anomaly', such as that described in the results of a recent qualitative study [13]. This qualitative study aimed to improve understanding of the lived experience of dissociation by interviewing 12 people with psychosis diagnoses who reported co-morbid dissociative experiences. The results of the study indicated that dissociation is commonly experienced as a subjective 'felt sense' that something is 'wrong', 'off', 'odd', or somehow anomalous. These sensations grouped into themes describing a type of anomaly, including 'strange', 'unreal' or

**Table 2. Summarising the results of N = 77 studies which carried out factor analysis on measures of dissociation or closely-related concepts (e.g. depersonalisation).**

| Reference | Factors | | | | Sample characteristics |
|---|---|---|---|---|---|
| **Dissociative Experiences Scale (Bernstein & Putnam, 1986):** | | | | | |
| Allen, Coyne & Console (1997) | Detachment from one's own actions | | Detachment from the self and the environment | | n = 266 female inpatients DES mean = 35.1 (SD = 23.2) |
| Amdur & Liberzon (1996) | Depersonalization / derealization | Memory disturbance | Absorption | Distractibility | n = 129 male patients DES mean = 30.43 (SD = 17.94) |
| Armour, Contractor, Palmieri & Elhai (2014) | Absorption | | Amnesia | Depersonalization / derealization | n = 165 university students DES mean not stated |
| Brunner, Parzer, Schmitt & Resch (2004) *German version* | Dissociative amnesia | | Absorption / imaginative involvement | Depersonalisation / derealization | n = 52 patients, 1056 control DES mean = 2.81 (SD = 1.67) (BPD); 1.40 (SD = 1.06) (schizophrenia); 1.72 (SD = 1.13) (controls) |
| Carleton, Abram & Asmundson (2010) *DES items & Tellegen Absorption Scale items* | Imaginative involvement | | Dissociative amnesia | Attentional dissociation | n = 841 undergraduates, 635 community women DES mean not stated |
| Darves-Bornoz, Degiovanni & Gaillard (1999) *French version* | Depersonalisation / derealisation | | Amnestic fragmentation of identity | Absorption-imaginative involvement | n = 140 victims of rape DES mean = 24.1 (SD = 16.5) |
| Dunn, Ryan & Paolo (1994) | Depersonalization / derealization | Moderate amnesiac dissociation | Absorption-imaginative involvement | Severe amnesiac dissociation | n = 493 male substance use patients DES mean not stated |
| Espirito Santo & Abreu (2009) *Portuguese version* | Depersonalization-Derealization | Absorption | Distractibility | Memory disturbances | n = 570 mixed patient & general population DES mean = 18.81 (SD = 13.82) |
| Fischer & Elnitsky (1990) *Including the Perceptual Alteration Scale items* | Single factor | | | | n = 507 undergraduates DES mean not stated |
| Holtgraves & Stockdale (1997) | Single factor | | | | n = 201 (study 1) & 195 (study 2) undergraduates DES mean not stated |
| Korlin, Edman & Nyback (2007) | Single factor | | | | n = 342 general population; 181 patients DES mean not stated |

(*Continued*)

**Table 2.** (Continued)

| Reference | Factors | | | | | | Sample characteristics |
|---|---|---|---|---|---|---|---|
| **Dissociative Experiences Scale (Bernstein & Putnam, 1986) cont'd:** | | | | | | | |
| Laroi, Billieux, Defeldre, Ceschi & van der Linden (2013) *French version* | Automatic pilot related dissociation | | | Defensive dissociation | | | n = 188 (study 1) & 210 (study 2) university students DES mean not stated |
| Lipsanen, Saarijarvi & Lauerma (2003) *Finnish version* | Single factor | | | | | | n = 924 general population DES mean = 8.41 (SD not stated) |
| Mazzotti et al. (2016) | Absorption | | Compartmentalization | | Detachment | | n = 780 patients; 2303 undergraduates and non-psychiatry patients DES mean = 14.63 (SD = 11.78) (general population); 20.02 (SD = 16.29) (psychiatry patients) |
| Olsen, Clapp, Parra & Beck (2013) | Absorption | | | Depersonalization | | | n = 575 (study 1) & 459 (study 2) female undergraduates DES mean not stated |
| Ray & Faith (1995) | Absorption / derealization | | Depersonalization | Segment amnesia | In situ amnesia | | n = 1190 undergraduates DES mean = 67.97 (SE = 1.03) (altered response format) |
| Ray, June, Turaj & Lundy (1992) *Revised version* | Fantasy / absorption | Segment amnesia | Depersonalization | In situ amnesia | Different selves | Denial | n = 264 university students DES mean not stated |
| Ross, Ellason & Anderson (1995) | Absorption / imaginative involvement | | Activities of dissociated states | Depersonalization / derealization | | | n = 274 patients with DID *Full text not available to authors* |
| Ross, Joshi & Currie (1991) | Absorption / imaginative involvement | | Activities of dissociated states | Depersonalization / derealization | | | n = 1055 general population DES mean = 10.8 (SD = 10.1) |
| Ruiz, Poythress, Lilienfeld & Douglas (2008) | Absorption | | Depersonalization | | Amnesia | | n = 1551 offenders DES mean = 18.6 (SD = 13.6) |
| Sanders & Green (1994) | Imaginative involvement | | Depersonalization / derealization | | Amnesia | | n = 860 undergraduates *Full text not available to authors* |

(*Continued*)

**Table 2.** (*Continued*)

| Reference | Factors | | | | Sample characteristics |
|---|---|---|---|---|---|
| Schimmenti (2016a) *Italian version* | Single factor | | | | n = 794 general population DES mean = 18.60 (SD = 13.85) |
| Schwartz & Frischholz (1991) | Amnestic dissociation | Absorption & imaginative involvement | Depersonalisation / derealization | | *Full text unavailable to authors* |
| **Dissociative Experiences Scale (Bernstein & Putnam, 1986) cont'd:** | | | | | |
| Simeon et al (1998) | Absorption | Amnesia | Depersonalization / derealization | | n = 50 patients with DPD; 20 controls DES mean = 23.41 (SD = 13.63) (DPD); 4.02 (SD = 2.91) (controls) |
| Soffer-Dudek, Lassri, Soffer-Dudek & Shahar (2015) | Absorption / imaginative involvement | Depersonalization / derealization | Amnesia | | n = 679 undergraduates DES mean not stated |
| Stockdale, Gridley, Balogh & Holtgraves (2002) | Absorption | Depersonalization | Amnesia | | n = 971 undergraduates DES mean not stated |
| Wright & Loftus (1999) *Standard, verbal, & comparative versions* | Single factor | | | | n = 75 undergraduates DES mean = 12.73 (SD = 2.39) |
| Zingrone & Alvarado (2001) | Single factor | | | | n = 308 university students DES mean = 21.70 (SD = 12.87) |
| **Adolescent Dissociative Experiences Scale (Armstrong, Putnam, Carlson, Libero & Smith 1997)** | | | | | |
| Armstrong et al. (1997) | Amnesia | Absorption | Passive influence | Depersonalization / derealization | n = 102 referred for psychological evaluation A-DES mean = 4.85 (SD = 1.14) (dissociative disorders) |
| De Pasquale, Sciacca & Hichy (2016) *Italian version* | Dissociative amnesia | Absorption & imaginative involvement | Depersonalisation / derealization | Passive influence | n = 633 students A-DES mean = 2.02 (SD = 1.47) |
| Farrington, Waller, Smerden & Faupel (2001) | Single factor | | | | n = 810 students A-DES mean = 2.66 (SD = 1.81) |
| Kerig et al (2016) | Depersonalization / derealization | Amnesia | Loss of conscious control | | n = 784 in juvenile detention A-DES mean = 58.07 (SD = 48.69) |

(*Continued*)

**Table 2.** (Continued)

| Reference | Factors | | | Sample characteristics |
|-----------|---------|---|---|------------------------|
| Muris, Merckelbach & Peeters (2003) | Single factor | | | n = 331 students A-DES mean = 1.27 (SD = 1.18) |
| Nilsson & Svedin (2006a) *Swedish version* | Single factor | | | n = 400 students; 20 outpatients A-DES mean = 0.84 (SD = 1.05) (non-clinical); 3.28 (SD = 1.89) (clinical) |
| Schimmenti (2016b) *Italian version* | Single factor | | | n = 1806 students A-DES mean = 1.92 (SD = 1.43) |
| Yoshizumi, Hamada, Kaida, Gotow & Murase (2010) *Japanese version* | Depersonalization | Disintegration of conscious control | Amnesia | n = 2272 students A-DES mean = 2.21 (SD = 1.69) |
| **Peritraumatic Dissociative Experiences Questionnaire (Marmar, Weiss & Metzler, 1997)** | | | | |
| Birmes et al. (2005) *French version* | Single factor | | | n = 48 (group 1); 43 (group 2) emergency department patients (critical incident victims) DES mean not stated |
| Boelen, Keijsers & van den Hout (2012) | Single factor | | | n = 168 grief processes research programme participants DES mean not stated |
| Brooks et al. (2009) | Altered awareness | | Derealization | n = 247 patients at trauma hospitals DES mean not stated |
| Bui et al. (2011) *Child version* | Single factor | | | n = 133 child emergency department patients DES mean not stated |
| Marshall, Orlando, Jaycox, Foy & Belzberg (2002) *Modified version* | Single factor | | | n = 284 youth exposed to community violence DES mean not stated |

(*Continued*)

**Table 2.**  (Continued)

| Reference | Factors | | | | Sample characteristics |
|---|---|---|---|---|---|
| Sijbrandij et al. (2012) | Altered awareness | | Derealization | | n = 219 police officers; 343 trauma-exposed civilians DES mean not stated |
| **Cambridge Depersonalisation Scale (Sierra & Berrios, 2000)** | | | | | |
| Aponte-Soto, Velez-Pastrana, Martinez-Taboas & Gonzalez (2014) | Anomalous body experience | Emotional and sensory numbing | Alienation from surroundings | Perceptual alterations | n = 300 general population DES mean = 13.20 (SD = 14.19) |
| Blevins, Witte & Weathers (2013) | Unreality and detachment | | Emotional and physical numbing | | n = 534 undergraduates DES mean not stated |
| Fagioli et al. (2015) *Italian version* | Detachment from the Self | Anomalous bodily experiences | Numbing | Temporal blunting | n = 149 inpatients & outpatients DES mean not stated |
| Sierra, Baker, Medford & David (2005) | Anomalous body experience | Emotional numbing | Anomalous subjective recall | Alienation from surroundings | n = 150 depersonalisation patients DES mean = 24.1 (SD = 14.7) |
| **Somatoform Dissociation Questionnaire (Nijenhuis, Spinhoven, van Dyck, van der Hart & Vanderlinden, 1996)** | | | | | |
| El-Hage, Darvez-Bornoz, Allilaire & Gaillard (2002) *French version* | Sensory neglect | | Subjective reactions to perceptive distortions | Vigilance modulation disturbance | n = 140 outpatients DES mean = 14.6 (SD = 12.9) |
| Mueller-Pfeiffer et al. (2010) *German version* | Single factor | | | | n = 225 psychiatry patients DES mean = 4.5 (SD = 2.6) (non-dissociative group); 32.9 (SD = 15.8) (dissociative group) |
| Nijenhuis et al. (1996) | Single factor | | | | n = 100 outpatients DES mean not stated |
| Nijenhuis, Spinhoven, van Dyck, van der Hart & Vanderlinden (1998) | Single factor | | | | n = 31 outpatients with dissociative symptoms DES mean not stated |
| **Multidimensional Inventory of Dissociation (Dell, 2006)** | | | | | |
| Dell (2013) | Discovering dissociated actions | | Lapses of recent memory and skills | Gaps in remote memory | n = 2569 clinical & non-clinical DES mean not stated |
| Dell (2006) | Single factor | | | | n = 817 (multiple groups) DES mean not stated |

*(Continued)*

**Table 2.** (Continued)

| Reference | Factors | | | | | Sample characteristics |
|---|---|---|---|---|---|---|
| Somer & Dell (2005) *Hebrew version* | Single factor | | | | | n = 151 undergraduate & general population DES mean not stated |
| **Curious Experiences Survey (Goldberg, 1999)** | | | | | | |
| Cann & Harris (2003) | Absorption | | Depersonalization | Amnesia | | n = 194 undergraduates DES mean not stated |
| Goldberg (1999) | Broad factor: Dissociation (31 items) | Subscale: dissociation (17 items) | Subscale: depersonalisation | Subscale: absorption | Subscale: amnesia | n = 755 general population DES mean not stated |
| **Dissociation Questionnaire (Vanderlinden, Van Dyck, Vandereycken, Vertommen & Verkes, 1993)** | | | | | | |
| Vanderlinden et al. (1993) | Identity confusion | Loss of control over behaviour, thoughts & emotions | Amnesia | | Absorption | n = 98 eating disorder patients DES mean not stated |
| Nilsson & Svedin (2006b) *Swedish version* | Identity confusion | Loss of control | Amnesia | | Absorption | n = 74 outpatient adolescents; 400 control adolescents DES mean not stated |
| **Perceptual Alterations Scale (Sanders, 1986)** | | | | | | |
| Sanders (1986) | Modification of Affect | | Modification of Control | Modification of Cognition | | *Full text not available to the authors* |
| Fischer & Elnitsky (1990) | Single factor | | | | | n = 507 undergraduates DES mean not stated |
| **Questionnaire of Experiences of Dissociation (Riley, 1988)** | | | | | | |
| Ray & Faith (1995) | Depersonalization | Process amnesia | Fantasy / daydream | Dissociated body behaviour | Trance | n = 1190 undergraduates DES mean = 67.97 (SE = 1.03) (altered response format) |
| Ray et al. (1992) *Revised version* | Depersonalization | Process amnesia | Fantasy / daydream | Dissociated body behaviour | Trance | n = 264 undergraduates DES mean = 2.17 (SE = 0.03) |
| **Scale of Bodily Connection (Price & Adams Thompson, 2007)** | | | | | | |
| Price & Adams Thompson (2007) | Body awareness | | | Body dissociation | | n = 291 undergraduates DES mean not stated |
| Price, Adams Thompson & Chieh Cheng (2017) | Body awareness | | | Body dissociation | | n = 3634 (various groups) DES mean not stated |

(*Continued*)

**Table 2.** (Continued)

| Reference | Factors | | | | | | Sample characteristics |
|---|---|---|---|---|---|---|---|
| **Clinician-Administered Dissociative States Scale** | Bremner et al. (1998) | Amnesia | Depersonalisation | Derealization | | | n = 68 PTSD patients DES mean not stated |
| **Dissociative Symptoms Scale** | Carlson et al. (2018) | Depersonalization / Derealization | Gaps | Sensory Misperceptions | Cognitive-Behavioural Reexperiencing | | n = 1592 multiple groups DES mean not stated |
| **The Dissociative Experiences Measure, Oxford** | Černis, Cooper & Chan (2018) | Unreality | Numb & Disconnected | Memory Blanks | Zoned Out | Vivid Internal World | n = 691 general population DES mean not stated |
| **Self-Experience Lifetime Frequency Scale** | Heering et al. (2016) | Disturbance of self-awareness | (Milder forms of) diminished self-affection or depersonalisation | | | | n = 426 psychosis patients; 526 healthy siblings; 297 healthy controls DES mean not stated |
| **Depersonalization scale** *including 12 items of Dixon's (1963) scale* | Jacobs & Bovasso (1992) | Inauthenticity | Self-negation | Self-objectification | Derealization | | n = 368 undergraduates DES mean not stated |

| Scale | Reference | Factors | | | | | Sample characteristics |
|---|---|---|---|---|---|---|---|
| **Wessex Dissociation Scale** | Kennedy et al. (2004) | Stage 1 (hallucinations / pseudo-hallucinations) | Stage 2 (including cognitive blanking, intrusions, numbing of affect) | Somatic dissociation | | | n = 80 psychology services patients; 80 undergraduates DES mean = 20.7 (SD = 16.2) clinical; 9.77 (SD = 7.68) non-clinical |
| **State Scale of Dissociation** | Kruger & Mace (2002) | Identity confusion, derealization, depersonalization | Conversion | Amnesia | Identity alteration | Hypermnesia | n = 67 patients; 63 controls DES mean not stated |
| **Traumatic Dissociation & Grief Scale** | Laor et al. (2002) | Perceptual distortions | Body-self distortions | Irritability | Guilt & anhedonia | | n = 303 children (202 displaced by earthquake; 101 not directly affected) DES mean not stated |
| **Trait Dissociation Questionnaire** | Murray, Ehlers & Mayou (2002) | Lability of mood & impulsivity · Sense of split self · Detachment from others & the world · Emotional numbing · Confusion & altered time senses · Amnesia for important life events · Memory lapses | | | | | n = 27 inpatient & 439 outpatient accident & emergency department patients 31.6% inpatients & 28.3% outpatients DES mean not stated |

(*Continued*)

**Table 2.** (Continued)

| Reference | Factors | | | | | Sample characteristics |
|---|---|---|---|---|---|---|
| *Scale unknown*: 'Questionnaire responses from 189 victims of life-threatening accidents' | Noyes & Slymen (1979) | Depersonalization | | Hyperalertness | Mystical consciousness | *Full text not available to authors* |
| **Steinberg dissociation questionnaires** (5 measures) (Steinberg & Schnall, 2000) | Sar, Alioğlu & Akyuz (2017) | Cognitive-emotional self-detachment | Perceptual detachment | Bodily self-detachment | Detachment from reality | n = 1301 undergraduates DES mean not stated |
| **Conversion Disorder Scale for Children** | Sarfraz & Ijaz (2014) | Feeling of disability | | Body pain | Seizures | n = 107 outpatients & controls *Full text not available to authors* |
| **Dissociation Tension Scale** | Stiglmayr et al. (2010) | Single factor | | | | n = 294 psychiatry patients DES mean not stated |
| **Child Dissociative Checklist** (Putnam, Helmers & Trickett, 1993) | Wherry, Neil & Taylor (2009) | Variability | | General externalising problems | Pathological dissociation | n = 232 children with abuse histories DES mean not stated |
| **Dissociative Symptoms Severity Scale–Child form** | Yalin Sapmaz et al. (2017) | Single factor | | | | n = 30 adolescent patients; 83 controls A-DES mean = 122.30 (SD = 52.61) (clinical); 65.96 (SD = 53.52) (controls) |

NB: References can be found in S1.

'disconnected' and could occur in relation to external or internal stimuli. This was defined as 'a felt sense of anomaly' (FSA).

Inspection of the above dissociation measures revealed that many items refer to experiences as 'different', 'altered', or otherwise suggest that the respondent has noticed changes from what they might have expected (e.g. 'some people have the experience of looking in a mirror and not recognizing themselves'; DES-II; [11]). As a result, we considered that there was adequate basis in the measures found in the systematic review to consider FSA as a phenomenological constant in many common dissociative experiences.

Whilst examining the measures in Table 2 for FSA, it became clear that there were further 'types' of FSA and a broader range of ways in which these could be experienced than those found by Černis, Freeman and Ehlers [13]. We therefore developed a theoretical framework for conceptualising a subset of 'FSA-type' dissociation where different 'domains' can be affected by a 'type' of anomaly. The 'domain' affected by FSA may be that of physical sensation, perception (of external or internal stimuli), mental content or processes (such as memory), or

in the experience of selfhood. The 'type' of anomaly may take the form of: unfamiliarity, unreality, automaticity or lack of control (where this would be unexpected), or unanticipated sense of detachment or absence.

This framework is summarised in an 'FSA matrix' (see Table 3), where each cell constitutes an experience where a domain is affected by a type of anomaly. For example, one's mind [domain] could be experienced as detached [type]–as in reports of being unable to easily access one's memories; or one's self [domain] may feel unfamiliar [type], such as in depersonalisation. In this way, the core experience of FSA unites these disparate experiences–all of which have previously been described as dissociative. The matrix in the format 'domain x type' enables the identification of which experiences may be included in this subset of dissociative experiences.

This conceptual framework was used to systematically generate items for a new measure; the development of which, in turn, empirically tests the proposed framework. The key aim of the empirical work reported in this paper is to develop a measure of FSA-type dissociation, using possibly the largest ever sample size for the development of a measure of dissociation or related constructs.

## Part 1: Developing the measure

First, the experience statements systematically generated using the FSA matrix were used as an item pool for generating a measure of FSA. Measure development took place within a non-clinical (general population) group.

**Table 3. The 'FSA matrix' used to systematically generate items for the development of a novel dissociation measure focusing on felt sense of anomaly, with one example shown per cell.**

| | | Types of Anomaly | | | | |
| --- | --- | --- | --- | --- | --- | --- |
| | | **Unreal** | **Unfamiliar** | **Automatic** | **Disconnected** | **Absent** |
| **Domains** | Mind | My thoughts don't seem real. | Some of the things in my head don't seem to be mine. | I can't access my thoughts or memories at will. | I feel detached from my own mind. | My mind goes completely empty. |
| | Affect | My emotions don't seem real | I have emotions that don't feel like they're mine. | My emotional reactions don't fit with the situation I am in. | I feel disconnected from my emotions. | I can't feel emotions. |
| | Physiology | My body (or parts of it) feels unreal or strange. | My body (or parts of it) feels like it doesn't belong to me. | My body (or parts of it) feels like it has a mind of its own. | I feel disconnected from the sensations in my body. | My body feels numb. |
| | Perception | The things happening around me seem unreal to me–like a dream or a movie. | One or more of my senses seem strange, distorted, or odd to me. | My sense of sight, touch, hearing (etc.) don't respond to me. | I feel as if I'm experiencing life from very far away. | I don't notice how much time passes. |
| | Identity | I feel that I'm not a real person. | I don't recognize myself. | I act like someone else without meaning to. | I feel disconnected from who I really am. | I feel like I don't exist. |
| | Behaviour | My actions feel fake or unreal. | Things I've done many times before seem new or unfamiliar. | I feel like I'm on automatic pilot. | I feel disconnected from my own actions. | I freeze, unable to do anything. |
| | World | The world around me seems unreal. | Places that I know seem unfamiliar. | - | I feel that I'm not part of the world around me. | I am absorbed in my own world and do not notice what is happening around me. |
| | Others | Other people seem unreal. | People I know seem unfamiliar. | - | I feel detached from the people I am close to. | Other people stop existing when I can't see them. |
| | | Unreal | Unfamiliar | Automatic | Disconnected | Absent |

(NB: Two cells (Automatic x World; Automatic x Others) are blank, as it would not be considered anomalous if these did not respond to a person's attempts at control.).

## Methods

**Study design.** The study was a questionnaire development study using an online cross-sectional self-report survey. A subsample of respondents also provided test-retest data for the novel questionnaire by completing the new measure twice more (Week 1 and Week 2).

**Ethical approval.** The study received ethical approval from the Central University Research Ethics Committee of the University of Oxford (ref: R57488/RE002).

**Participants.** Participants were recruited via social media, the majority via Facebook Ads. The advertisements were titled "*Mapping dissociation in mental health*" and stated that questionnaires concerned "common thoughts and feelings". The information sheet described dissociation as "*strange feelings and experiences such as 'spacing out', feeling 'unreal', or feeling detached from the world around you*". Inclusion criteria were deliberately very broad: any adult (age 18 years or over) normally resident in the UK. There were no exclusion criteria, and no required level of current or past dissociation. Due to the online survey format, it was not possible to directly assess capacity to consent. However, this was assumed since the participant was required to open the survey hyperlink, read the information sheet, and complete the consent statements independently. Upon declining to consent, the survey was not shown and the end page with resources for further support was instead displayed.

13186 responses were recorded by Qualtrics [14]. 144 (1.09%) did not consent to the study, and 307 (2.33%) indicated consent but then left the survey without continuing onto the first page of measures. After removing participants who did not meet the inclusion criteria, or had high levels of missing data (greater than 20% in any of the measures required for analysis), the final sample was 8861. The characteristics of the sample can be found in Table 4.

**Procedures.** Questionnaires were completed online using Qualtrics. Therefore, informed consent and assessment were both carried out online. The questionnaire landing page contained the participant information sheet and statements regarding informed consent, as per the British Psychological Society guidelines for ethical internet-mediated research [15]. Participants were told that the aim of the study was to explore dissociation and common thoughts, feelings, and experiences, and that they need not have experienced dissociation in order to take part. After acknowledging the consent statements, participants were asked the demographic questions, and shown the item pool and measures described below (see *Measures*).

**Table 4. Summarising the descriptive statistics for the three subsamples used for measure development.**

| | **Sample 1** (n = 2953) | **Sample 2** (n = 2954) | **Sample 3** (n = 2954) |
|---|---|---|---|
| Gender | 287 (9.7%) male | 317 (10.7%) male | 280 (9.5%) male |
| | 2557 (86.6%) female | 2544 (86.1%) female | 2568 (86.9%) female |
| | 80 (2.7%) other | 75 (2.5%) other | 78 (2.6%) other |
| Ethnicity | 2751 (93.1%) White | 2751 (93.1%) White | 2768 (93.7%) White |
| "Have you ever experienced mental health difficulties?" | 2528 (85.6%) Yes | 2497 (84.5%) Yes | 2508 (84.9%) Yes |
| | 360 (12.2%) No | 405 (13.7%) No | 388 (13.1%) No |
| "If yes, are these still ongoing?" | 1929 (65.3%) Yes | 1900 (64.3%) Yes | 1943 (65.8%) Yes |
| | 534 (18.1%) No | 537 (18.2%) No | 519 (17.6%) No |
| | **Range Mean (SD)** | | |
| Age | 18–88 40.04 (15.67) | 18–84 40.02 (15.84) | 18–85 40.38 (15.78) |
| | **Mean (SD)** | **Mean (SD)** | **Mean (SD)** |
| Dissociative Experiences Scale (DES)* | 2.37 (1.85) | 2.41 (1.89) | 2.40 (1.89) |
| PTSD Checklist (PCL-5)* | 30.07 (20.14) | 29.29 (20.22) | 27.00 (19.93) |

*t-tests for differences in mean scores between genders male and female found no significant statistical differences in any sample.

The survey was accessible on desktop and mobile web browsers. Incomplete datasets were retrieved automatically after a week of non-activity and added to the dataset.

Data collection began on May 24, 2018 and ended on July 23, 2018. Test-retest data were collected between September 3 and 13, 2018.

**Measures.**     *Černis Felt Sense of Anomaly Scale (ČEFSA)*. First, an initial item pool of 98 items was systematically generated by EČ, DF and AE by completing the cells of the aforementioned FSA matrix (Table 3). For example, the cell at the juncture of affect [domain] and unreal [type] would produce the item "my emotions don't seem real". Using this method, a minimum of two items per cell were generated (with the exception of 'world x automatic' and 'others x automatic' where it was considered that it would not be anomalous to experience the world or others as not under one's control). Generated items were required to clearly relate to both the domain and the type of anomaly. Further, they were not to describe a reaction or behaviour (as these may be idiosyncratic, and are not dissociative phenomena in their own right), nor could items be written such that the item might have surface validity for another disorder (in order to minimise misinterpretation by respondents). Items were validated against these criteria via discussion between EČ, DF and AE.

Additionally, six items were generated that were 'global', in that they only described FSA without reference to specific domain or type (e.g. 'I feel odd', 'Things seem strange'; see S2). These items were generated to develop a supplementary brief "Global FSA" scale (see *Statistical analysis*).

All 104 items were checked for readability by volunteers with lived experience of mental health problems. In particular, volunteers checked that it was clear to a layperson what the items were asking, and that the language used was easily accessible throughout.

Items were rated for the past two weeks on a Likert scale from "0 Never" to "4 Always", with the instruction 'Please read the following items and rate how often you have experienced these over the past TWO WEEKS'.

*Dissociative Experiences Scale II (DES-II;* [11]*)*. The DES-II comprises 28 items each rated from 0% to 100%. Items cover dissociative and amnestic experiences such as "Some people sometimes find that they are approached by people that they do not know, who call them by another name or insist that they have met them before." Higher scores indicate greater dissociation. No time period is specified in the instructions.

*Post-Traumatic Symptom Disorder Checklist (PCL-5;* [16]*)*. To assess PTSD symptoms over the past month, the PCL-5 contains 20 items such as "feeling very upset when something reminded you of the stressful experience", rated on a five-point Likert scale from "0 not at all" to "4 extremely". Participants were asked to rate "the most upsetting event" they had experienced, indicated via selecting from a list including "end of a relationship", "natural death of a significant other", "severe accident", and "other not listed". Higher scores indicate greater trauma symptomatology.

**Statistical analysis.**     Analyses were conducted in R, version 3.5.1 [17] with packages psych [18] and lavaan (version 0.6–3; [19]). For analysis, the sample was split into three equal subsamples of nearly 3000 people. This was to enable refinement of the item pool via two exploratory factor analyses with appropriately large samples, and then a test of the factor structure in a third subsample via confirmatory factor analysis. Sample splitting was done by randomly allocating cases to subsets using a function in R.

The global items were separated from items developed using the FSA matrix and analysed separately. This was done for two reasons: first, because the construct underlying these items was distinct (they represent general FSA, rather than an interaction between a type and domain); and second, to fulfil the aim of providing a very brief, standalone tool with which to measure the underlying common denominator of FSA.

Following measure development and confirmatory factor analysis, the psychometric properties of the final scale(s) were assessed. Validity was tested via convergent validity with an existing dissociation measure (the DES-II) using Pearson correlation. Further, confirmatory factor analyses were carried out to test the factor structure in participants scoring above and below the clinical cut-off on the PTSD measure (PCL-5; [16]). Reliability was assessed via internal consistency (Cronbach's alpha) and one-week test-retest reliability (intra-class correlation).

## Results

Each of the three subsamples had a mean age of 40 years, scored within the general population range [11] on the DES, and highly on the PCL-5 (see Table 4). In each sample, approximately 86% of respondents were female, 93% were White, and 85% reported lifetime mental health difficulties (with a further two thirds of these reporting that such experiences are ongoing).

**Items developed from the FSA matrix: The Černis Felt Sense of Anomaly scale.** Exploratory Factor Analysis (EFA) with oblique rotation was carried out on the first two subsamples, with items that loaded weakly to a factor (less than 0.3) or cross-loaded strongly across multiple factors (loadings for different factors within 0.2 of each other) discarded after each EFA. The first EFA (n = 2953) indicated that a seven-factor solution was the most appropriate using parallel analysis and model comparison tests ($X^2$ (4088): 20333.396, p<0.001, CFI = 0.922, TLI = 0.909, RMSEA = 0.037, SRMR = 0.018). Factors were identified as 'Anomalous Experience of the Self', 'Anomalous Experience of the Physical Body', 'Altered Sense of Familiarity', 'Anomalous Experience of Emotion', 'Altered Sense of Connection', 'Altered Sense of Agency', and 'Altered Sense of Reality'. After the second EFA (n = 2954), only five items meeting the aforementioned criteria were retained per factor. These were selected based on which combination of five items produced a theoretically well-rounded set of items (i.e. not all asking about the same experience). This was achieved via consensus between EČ, DF and AE. The result was a measure of 35 items, each of which load strongly to their factor ($X^2$ (2138) = 10215.014, p<0.001, CFI = 0.944, TLI = 0.931, RMSEA = 0.036, SRMR = 0.016). The final scale (the Černis Felt Sense of Anomaly; ČEFSA scale) can be found in S 2.

On the third and final subsample, a Confirmatory Factor Analysis (CFA) (n = 2954) was carried out to test the seven-factor structure of the 35-item measure. This showed a good model fit for a second-order factor structure ($X^2$ (553) = 4989.435, p<0.001, CFI = 0.929, TLI = 0.924, RMSEA = 0.052, SRMR = 0.047), where the high loadings of each of the seven factors indicate that they well-represent the higher-order construct of FSA-type dissociation (Fig 2).

The ČEFSA showed good psychometric properties (Table 5). There was good convergent validity with the DES-II (r = 0.802, p<0.001), and excellent test-retest reliability over a week (ICC = 0.92; 95% CI = 0.88–0.94; p<0.001). Internal consistency within the seven subscales was excellent (Cronbach's alphas of 0.86 to 0.92).

Further, CFAs were carried out after dividing cases in the sample with less than 20% missing data for ČEFSA items and the PCL-5 (Weathers et al., 2013) (N = 7021) into two groups: those scoring above (N = 2836), and those below (N = 4135) the clinical cut off of 33 on the PCL-5 (above group: mean = 50.38, SD = 11.07; below group: mean = 15.33, SD = 9.86). Both demonstrated a good model fit, indicating that the factor structure of the ČEFSA is robust even in a population with clinically significant trauma symptoms (high: $X^2$ (553) = 4758.673, p<0.001, CFI = 0.913, TLI = 0.906, RMSEA = 0.052, SRMR = 0.054; low: $X^2$ (553) = 5487.204, p<0.001, CFI = 0.919, TLI = 0.913, RMSEA = 0.046, SRMR = 0.050).

**Global FSA items: The Global Felt Sense of Anomaly scale.** The same methodology was followed to separately develop and validate the Global FSA Scale: EFA with oblique rotation in

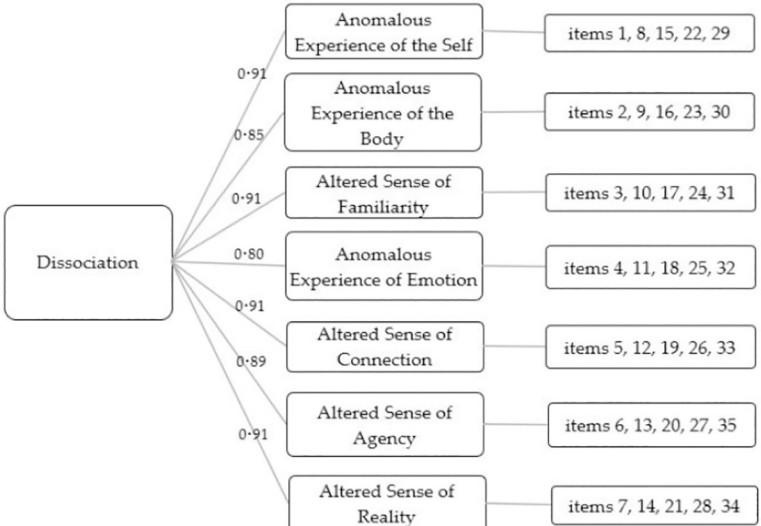

**Fig 2. The second-order seven-factor structure of the Černis Felt Sense of Anomaly measure, with factor loadings onto the latent variable (dissociation).**

the first and second subsamples indicated a single factor structure (1st EFA: $X^2$ (9) = 275.050, p<0.001, CFI = 0.975, TLI = 0.958, RMSEA = 0.100, SRMR = 0.019; 2nd EFA: $X^2$ (9) = 301.402, p<0.001, CFI = 0.969, TLI = 0.949, RMSEA = 0.105, SRMR = 0.021). Following the second EFA, only five items were retained, following the same procedure as described for the main scale, above. Additionally, one item was reworded for clarity, and therefore the CFA was carried out in the test-retest subsample (n = 240), as these participants answered the newer version of the item. The CFA indicated that the one-factor structure with 5 items was a good model fit ($X^2$ (9) = 312.350, p<0.001, CFI = 0.970, TLI = 0.950, RMSEA = 0.107, SRMR = 0.021).

The Global FSA Scale was also found to have good psychometric properties (Table 5). Again, the scale demonstrated good convergent validity with the DES-II (r = 0.699, p<0.001), good test-retest reliability (ICC = 0.84; 95% CI = 0.78–0.89; p<0.001), and excellent internal consistency (Cronbach's alpha = 0.95).

**Relationship between measures.** Correlations were carried out between the Global FSA Scale and seven factors derived from the FSA matrix (Černis Felt Sense of Anomaly scale). These indicated a high level of correlation (Table 6).

Additionally, the internal consistency was high when the items of the main seven-factor scale and the Global FSA scale were analysed together (Cronbach's alpha = 0.98). This indicates that as well as being used independently as a 5-item 'screener' for FSA, the general items scale may potentially act as an optional 'eighth factor' when assessing FSA-type dissociation in full.

## Part 2: Validation in a clinical group

Next, the measure resulting from initial development in Part 1 was tested for psychometric fit in a clinical group. Whilst dissociation has been demonstrated to have associations with a broad range of mental health presentations [20], a group of patients with non-affective psychosis diagnoses were recruited to validate the new scale in a clinical group. 1038 people with psychosis diagnoses were surveyed as part of the *Exploring Unusual Feelings* study which aimed to explore the relationship between dissociation, psychotic symptoms, and other psychological factors. It is appropriate to study dissociation within the context of psychosis since dissociation is thought to be transdiagnostic [21], and to occur at an elevated level in psychosis diagnoses

**Table 5. Summarising the psychometric properties of the Černis Felt Sense of Anomaly (ČEFSA) scale, and the 5 global felt sense of anomaly items which can act as a standalone brief measure.**

| Psychometric | | Statistic |
|---|---|---|
| **Items developed from FSA matrix (35 items, 7 factors) (the Černis Felt Sense of Anomaly scale):** | | |
| **Test re-test reliability** (n = 240) | ICC statistic | 0.92 |
| | Lower bound | 0.88 |
| | Upper bound | 0.94 |
| | Degrees of freedom | 239; 239 |
| | K | 2 |
| | P | <0.001 |
| | F statistic | 25 |
| **Internal consistency** (n = 2954) | *Factor* | *Cronbach's alpha* |
| | Anomalous Experience of the Self | 0.87 |
| | Anomalous Experience of the Body | 0.91 |
| | Altered Sense of Familiarity | 0.90 |
| | Anomalous Experience of Emotion | 0.92 |
| | Altered Sense of Connection | 0.91 |
| | Altered Sense of Agency | 0.86 |
| | Altered Sense of Reality | 0.89 |
| | Total (35 items) | 0.97 |
| **Convergent validity** (n = 2954) (vs. DES-II) | Pearson's r | 0.802 |
| **Global Felt Sense Of Anomaly Scale (5 items, 1 factor):** | | |
| **Test re-test reliability** (n = 240) | ICC statistic | 0.84 |
| | Lower bound | 0.78 |
| | Upper bound | 0.89 |
| | Degrees of freedom | 239; 239 |
| | K | 2 |
| | P | <0.001 |
| | F statistic | 12 |
| **Internal consistency** (n = 240) | Cronbach's alpha | 0.95 |
| **Convergent validity** (n = 240) (vs. DES-II) | Pearson's r | 0.699 |

[22]. Further, as outlined in *Definition & Framework Development*, above, the concept of FSA has been established as relevant to this patient group in a qualitative study with 12 people with psychosis [13].

**Table 6. Summarising the correlation statistics (r) between the Global FSA scale and the factor scores and Černis Felt Sense of Anomaly (ČEFSA) scale total and factor scores.**

| Factor | r statistic |
|---|---|
| ČEFSA total score | 0.856 |
| Anomalous Experience of the Self | 0.797 |
| Anomalous Experience of the Body | 0.761 |
| Altered Sense of Familiarity | 0.767 |
| Anomalous Experience of Emotion | 0.674 |
| Altered Sense of Connection | 0.848 |
| Altered Sense of Agency | 0.682 |
| Altered Sense of Reality | 0.801 |

NB: All p values <0.001.

## Methods

**Study design.**    The design was a cross-sectional self-report questionnaire study.

**Ethical approval.**    The study received ethical approval from the NHS Health Research Authority, London (City & East) Research Ethics Committee (ref: 19/LO/1394).

**Procedure & participants.**    This study was supported by the National Institute of Health Research (NIHR) Clinical Research Network (CRN). Participants were recruited by CRN research assistants and clinical studies officers embedded in clinical teams and Research and Development departments across 36 NHS trusts. Research workers from these teams approached patients meeting the inclusion criteria, assessed capacity to consent, gained informed consent, and supported participants to complete the assessment pack. Inclusion criteria were broad: any person (age 16 years or over), currently under the care of an NHS mental health service, with a diagnosis of non-affective psychosis, who was willing and able to give informed consent to participate. Exclusion criteria were: insufficient English language to complete the questionnaires with support, and an affective psychosis diagnosis (i.e. psychotic depression, bipolar disorder).

Recruitment took place between 18th October 2019 and 19th March 2020. Datasets from 1038 participants were returned. For this analysis, only cases without high levels of missing data in the ČEFSA measure (less than or equal to 20% missing) were retained for analysis. This resulted in a participant group of 1031 patients for the ČEFSA validation, and 1028 for the Global FSA measure validation analysis.

In the ČEFSA validation group (n = 1031), the majority of participants were White (66.83%), male (69.74%), under the care of mental health services as an outpatient (74.30%) and had a diagnosis of Schizophrenia (64.60%). The mean age of the sample was 41.54 (SD = 12.32) years. See Table 7 for full demographic details.

**Table 7. Showing the demographic data for the clinical participant group (n = 1031).**

| Demographic | | n (% of group) |
|---|---|---|
| **Gender** | *Female*: <br> *Male*: <br> *Other*: | 303 (29.39%)719 (69.74%)5 (0.48%) |
| **Ethnicity** | *White (any)*: <br> *Black (any)*: <br> *Asian (any)*: <br> *Mixed / Multiple*: <br> *Other*: | 689 (66.83%) <br> 176 (17.07%) <br> 98 (9.51%) <br> 44 (4.27%) <br> 18 (1.75%) |
| **Diagnosis** | *Schizophrenia* <br> *Schizoaffective* <br> *Delusional Disorder* <br> *Psychotic Disorder NOS\** <br> *First Episode Psychosis* <br> *Other psychosis disorder* | 666 (64.60%) <br> 153 (14.84%) <br> 14 (1.36%) <br> 69 (6.69%) <br> 105 (10.18%) <br> 24 (2.33%) |
| **Care team type** | *Inpatient* <br> *Outpatient* <br> *Early intervention* | 265 (25.70%) <br> 766 (74.30%) <br> *124 (12.03%)* |
| **Demographic** | **Range** | **Mean (Standard Deviation)** |
| **Age** | 18–74 | 41.54 (12.32) |
| **Measure** | **Range** | **Mean (Standard Deviation)** |
| **Černis Felt Sense of Anomaly scale**\*\* | 0–140 | 39.54 (30.48) |

\*including Unspecified Non-Organic Psychosis.

\*\* \*t-tests for differences in mean scores between genders male and female found no significant statistical differences.

The Global FSA scale validation group (n = 1028) did not differ significantly from the ČEFSA validation group in terms of any demographics presented in Table 7. Their mean score on the Global FSA scale was 7.85 (SD = 5.61; range = 0–20).

**Measures.**   Participants completed the Černis Felt Sense of Anomaly (ČEFSA) and the Global FSA scales as developed in Part 1, above.

**Statistical analysis.**   Analyses were conducted in R, version 3.6.3 [17] with packages psych (version 1.9.12.31; [18]) and lavaan (version 0.6–5; [19]).

The measure model fit was assessed using Confirmatory Factor Analysis (CFA) with MLR robust maximum likelihood estimator in the clinical group (n = 1015). Due to restrictions within the study design, it was not possible to collect data for assessing convergent validity against another dissociation measure, nor test-retest reliability. Internal reliability was ana-lysed using Cronbach's alpha.

## Results

**Černis Felt Sense of Anomaly (ČEFSA) scale.**   Confirming that factor analysis was appro-priate, Bartlett's test of Sphericity was significant ($\chi^2$ = 4269.89, df = 595, $p<0.001$) and the Kaiser-Meyer-Olkin test of sampling adequacy was high (KMO = 0.98).

Confirmatory Factor Analysis (CFA) indicated an adequate fit for an 8-factor second-order model ($X^2$ (553) = 1623.641, $p<0.001$, CFI = 0.927, TLI = 0.921, RMSEA = 0.043, SRMR = 0.043), with factor loadings as shown in Table 8. In this group, the ČEFSA had good internal consistency (whole scale Cronbach's alpha = 0.97).

**Global FSA scale.**   Confirming that factor analysis was appropriate, Bartlett's test of Sphe-ricity was significant ($\chi^2$ = 684.543, df = 10, $p<0.001$) and the Kaiser-Meyer-Olkin test of sam-pling adequacy was adequate (KMO = 0.89).

CFA indicated an adequate fit for a 1-factor model ($X^2$ (5) = 12.127, $p = 0.033$, CFI = 0.996, TLI = 0.991, RMSEA = 0.037, SRMR = 0.011). In this group, the global FSA scale had good internal consistency (whole scale Cronbach's alpha = 0.92).

## Discussion

The aim of this paper is to demarcate a substantial subset of dissociative experiences using a data-driven approach. Since there continues to be controversy regarding the mechanisms of dissociation [6], we have taken the 'bottom-up' approach of focusing on the phenomenological level to achieve this. By so doing, we have demonstrated that a seemingly disparate set of

**Table 8.  Summarising the factor loadings and internal consistencies of the Černis Felt Sense of Anomaly scale.**

| Factor: | Factor loading onto the latent construct of dissociation | Internal consistency: *Cronbach's alpha* |
|---|---|---|
| Anomalous Experience of the Self | 0.96 | 0.83 |
| Anomalous Experience of the Body | 0.89 | 0.85 |
| Altered Sense of Familiarity | 0.92 | 0.84 |
| Anomalous Experience of Emotion | 0.78 | 0.89 |
| Altered Sense of Connection | 0.98 | 0.87 |
| Altered Sense of Agency | 0.96 | 0.84 |
| Altered Sense of Reality | 0.92 | 0.85 |
| **Whole scale** (35 items): | | 0.97 |

common dissociative experiences can be unified by the phenomenological common denominator of 'a felt sense of anomaly' (FSA).

The development of the ČEFSA (Černis Felt Sense of Anomaly) scale constitutes the first empirical test of the theoretical framework of the subset of 'FSA-type' dissociation outlined here. This framework posits that a set of common dissociative experiences take the form of a felt sense of anomaly, which may be of a particular 'type' (e.g. unfamiliarity, unreality) and may occur in a particular 'domain' of experience (e.g. physical body, external world). The second-order seven-factor solution of the ČEFSA closely follows the structure of the FSA matrix developed from this framework. Four factors of the ČEFSA (Altered Sense of Familiarity, of Connection, of Agency, and of Reality) reflect nearly all 'type' columns of the matrix. The remaining three factors of the ČEFSA (Anomalous Experience of the Self, of the Body, and of Emotion) reflect three of the eight 'domain' rows of the matrix–one might hypothesise that these are particularly important domains in the context of FSA-type dissociation.

Importantly, this scale may also be a valuable tool for the assessment of FSA-type dissociation. The ČEFSA is a novel measure of dissociative experiences which share a core feature of FSA, and is psychometrically robust, easy to read, and appropriate for both non-clinical respondents (including those reporting trauma symptoms) and clinical respondents with diagnoses of psychosis. The correlation between the ČEFSA and DES was high, likely because of the number of items within the DES that concern FSA. However, the ČEFSA has the additional benefit of being developed through a systematic delineation of the concept of FSA. Consequently, it reflects an underlying theoretical framework, and reflects this construct comprehensively. As a result, the ČEFSA includes less severe, or more difficult to articulate experiences that may not have received adequate attention previously such as 'I feel like I don't have a personality' and 'I can't feel emotions' in the Anomalous Experience of the Self and Anomalous Experience of Emotion factors.

Of course, it remains to be seen whether 'FSA-type' dissociative experiences represent a separable construct or type of dissociation with a shared aetiology. Whilst we envisage FSA-type dissociation as a set of experiences at the milder end of a dissociation spectrum (albeit causing considerable distress; [13]), it currently stands only as a working hypothesis, and requires thorough investigation. Specifically, further exploration of this construct and the factor structure of the corresponding measure within other clinical groups would be a logical and necessary next step for the development of the ideas proposed here, particularly as dissociation is considered transdiagnostic [21] and FSA-dissociation has recently been demonstrated to relate to a broad range of subclinical mental health presentations, including depression and anxiety as well as psychotic and post-traumatic symptoms [20].

Despite being a working hypothesis, we hope that the construct of FSA-type dissociation will prove useful in clinic and research because of its emphasis on the core lived experience of FSA. It is a strength of the present study that the proposed theoretical framework is consistent with first-person reports, and that the measure items were approved by experts by experience. Centring the framework on this core experience distils the surface-level complexity of such presentations into a broad but nevertheless descriptive heuristic which may aid recognition of such symptoms when they arise. It also enables clarity about which experiences are included in this subtype (for example, by using the FSA matrix), which is perhaps less straightforward with definitions which are built upon proposed mechanisms.

It is important to note that the construct of FSA-type dissociation proposed here does not preclude existing suggestions of dissociative subtypes. For example, domains relating to the self, the body and internal experiences also describe 'depersonalisation', and domains relating to the external world and other people describe 'derealisation'. There is also feasible overlap between Holmes et al.'s [7] detachment and the 'disconnected' (and possibly 'unreal' and

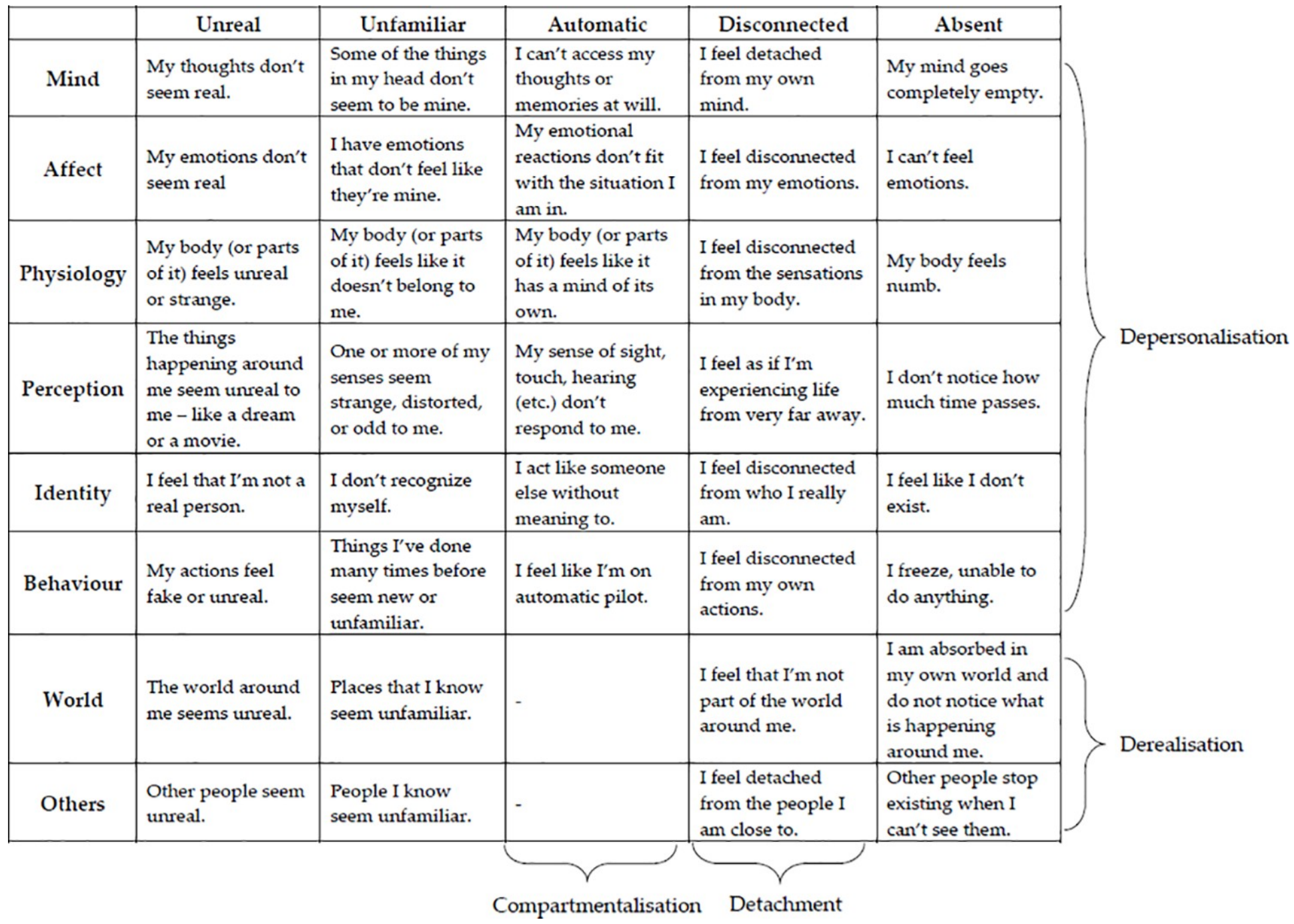

| | Unreal | Unfamiliar | Automatic | Disconnected | Absent |
|---|---|---|---|---|---|
| Mind | My thoughts don't seem real. | Some of the things in my head don't seem to be mine. | I can't access my thoughts or memories at will. | I feel detached from my own mind. | My mind goes completely empty. |
| Affect | My emotions don't seem real | I have emotions that don't feel like they're mine. | My emotional reactions don't fit with the situation I am in. | I feel disconnected from my emotions. | I can't feel emotions. |
| Physiology | My body (or parts of it) feels unreal or strange. | My body (or parts of it) feels like it doesn't belong to me. | My body (or parts of it) feels like it has a mind of its own. | I feel disconnected from the sensations in my body. | My body feels numb. |
| Perception | The things happening around me seem unreal to me – like a dream or a movie. | One or more of my senses seem strange, distorted, or odd to me. | My sense of sight, touch, hearing (etc.) don't respond to me. | I feel as if I'm experiencing life from very far away. | I don't notice how much time passes. |
| Identity | I feel that I'm not a real person. | I don't recognize myself. | I act like someone else without meaning to. | I feel disconnected from who I really am. | I feel like I don't exist. |
| Behaviour | My actions feel fake or unreal. | Things I've done many times before seem new or unfamiliar. | I feel like I'm on automatic pilot. | I feel disconnected from my own actions. | I freeze, unable to do anything. |
| World | The world around me seems unreal. | Places that I know seem unfamiliar. | - | I feel that I'm not part of the world around me. | I am absorbed in my own world and do not notice what is happening around me. |
| Others | Other people seem unreal. | People I know seem unfamiliar. | - | I feel detached from the people I am close to. | Other people stop existing when I can't see them. |



**Fig 3. The 'FSA matrix' with previous conceptualisations overlaid.** (NB: Detachment and compartmentalisation refer to constructs outlined by Holmes et al. [7]).


'unfamiliar') 'type' of FSA, and between compartmentalisation and the 'automatic' (and possibly 'absent') types (Fig 3). Accordingly, it would be of interest to explore this suggestion further using the ČEFSA and the recently published Detachment and Compartmentalization Inventory (DCI; [23]).

There are, of course, limitations to the proposed theoretical framework. One major criticism may be the omission of traditional 'dissociative amnesia' experiences from the FSA matrix. This symptom is considered a cardinal feature of dissociation, comprising a diagnostic entity in its own right [24], and forming a factor in many established dissociation measures (Table 2), including the DES [11]. Whilst detachment or unfamiliarity of memory falls within the framework of FSA-type dissociation, the relationship of FSA to frank dissociative amnesia (such that another 'part' of the personality retains a memory that is entirely inaccessible by another 'part') is unclear. Further exploration is required to determine whether such experiences may be described by the conjunction of 'absent' and 'mind' in the FSA-matrix, or whether a 'felt sense of anomaly' simply does not occur with dissociative amnesia in the same way as other items included in the ČEFSA scale. Indeed, an inherent feature of FSA is the subjective experience of (and plausibly, appraisal of) anomaly–however, many compartmentalisation symptoms are defined by a subjective *absence* or inaccessibility of experience until after the event has passed [7]. The ČEFSA scale therefore does not capture processes where the person completely loses awareness of their current surroundings or responds to content in memory as if it represented the present, and further research is required to determine the compatibility of the concept of FSA with these processes. However, we emphasise that FSA-type dissociation does not preclude the possibility of dissociative amnesia, and that the ČEFSA scale includes experiences where memory is experienced with a subjective sense of strangeness, including detachment and unfamiliarity.

A key limitation of the measure development is the sampling method in Part 1. Recruitment via Facebook ads attracted a sample which does not accurately reflect the general population, since it relies upon people who engage with social media and are willing to partake in online questionnaires. In particular, there is a large skew towards female gender and White ethnicity in the sample demographics, as well as a high level of self-reported mental health difficulties. This is further reflected in the relatively high group mean scores on the PTSD measure and high number of people exceeding the clinical cut-off score of 33, which suggests that this sample–although drawn from the general population–contains higher levels of post-traumatic stress than expected. People who have dissociative symptoms may also be overrepresented, likely resulting from self-selection bias due to the title of the study. Further, the quality of the data is unclear, as there is some evidence that up to eleven percent of Facebook profiles may be duplicates [25]. It is also a limitation of the study that test-retest data could not be collected in Part 2.

## Conclusions

This study defines a discrete set of common dissociative experiences unified by a phenomenological common denominator ('Felt Sense of Anomaly'; FSA), and demonstrates that the proposed framework underlying these experiences finds support in non-clinical (general population) and psychosis groups. The measure developed here is intended to support clinicians and researchers to detect this type of experience, which we hope will facilitate progress in the challenging field of dissociation more broadly.

## Supporting information

**S1 File.**
(DOCX)

## Acknowledgments

The authors would like to thank the R&D and NIHR CRN staff within the following NHS trusts for participating in the 'Exploring Unusual Feelings' study: Avon and Wiltshire Mental Health Partnership NHS Trust; Black Country Healthcare NHS Foundation Trust; Barnet, Enfield & Haringey Mental Health NHS Trust; Birmingham and Solihull Mental Health NHS Foundation Trust; Berkshire Healthcare NHS Foundation Trust; Birmingham Women's and Children's NHS Foundation Trust; Camden and Islington NHS Foundation Trust; Central and North West London NHS Foundation Trust; Coventry and Warwickshire Partnership NHS Trust; Cambridge and Peterborough NHS Foundation Trust; Cornwall Partnership NHS Foundation Trust; Cheshire and Wirral Partnership NHS Foundation Trust; Cumbria, Northumberland, Tyne and Wear NHS Foundation Trust; Dorset Healthcare University NHS Foundation Trust; Dudley and Walsall Mental Health Partnership NHS Trust; Devon Partnership NHS Trust; East London NHS Foundation Trust; Gloucestershire Health and Care NHS Foundation Trust; Hertfordshire Partnership University NHS Foundation Trust; Humber Teaching NHS Foundation Trust; Kent and Medway NHS and Social Care Partnership Trust; Leicestershire Partnership NHS Trust; Midlands Partnership NHS Foundation Trust; Mersey Care NHS Foundation Trust; North East London Foundation Trust; North Staffordshire Combined Healthcare NHS Trust; Oxford Health NHS Foundation Trust; Pennine Care NHS Foundation Trust; Surrey and Borders Partnership NHS Foundation Trust; Sheffield Health & Social Care NHS Foundation Trust; Solent NHS Trust; Somerset Partnership NHS Foundation Trust; Southern Health NHS Foundation Trust; Tees, Esk and Wear Valleys NHS Foundation Trust; Worcestershire Health and Care NHS Trust; and West London NHS Trust.

## Author Contributions

**Conceptualization:** Emma Černis, Anke Ehlers, Daniel Freeman.

**Data curation:** Emma Černis, Esther Beierl.

**Formal analysis:** Emma Černis, Esther Beierl.

**Funding acquisition:** Emma Černis.

**Investigation:** Emma Černis.

**Methodology:** Emma Černis, Anke Ehlers, Daniel Freeman.

**Project administration:** Emma Černis, Andrew Molodynski.

**Resources:** Andrew Molodynski.

**Software:** Esther Beierl.

**Supervision:** Anke Ehlers, Daniel Freeman.

**Visualization:** Emma Černis.

**Writing – original draft:** Emma Černis.

**Writing – review & editing:** Emma Černis, Esther Beierl, Andrew Molodynski, Anke Ehlers, Daniel Freeman.

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
