## [Decision Letter · Decision Letter 0]

9 Nov 2020

PONE-D-20-20041

A new perspective and assessment measure for common dissociative experiences: ‘Felt Sense of Anomaly’

PLOS ONE

Dear Dr. Černis,

Thank you for submitting your manuscript to PLOS ONE. After careful consideration, we feel that it has merit but does not fully meet PLOS ONE’s publication criteria as it currently stands. Therefore, we invite you to submit a revised version of the manuscript that addresses the points raised during the review process.

We look forward to receiving your revised manuscript.

Kind regards,

Vedat Sar, M.D.

Academic Editor

PLOS ONE

Journal Requirements:

3. Please describe in your methods section how capacity to consent was determined for the participants in this study.

Reviewers' comments:

Reviewer's Responses to Questions

**Comments to the Author**

1. Is the manuscript technically sound, and do the data support the conclusions?

Reviewer #1: Yes

Reviewer #2: Partly

2. Has the statistical analysis been performed appropriately and rigorously? 

Reviewer #1: N/A

Reviewer #2: I Don't Know

3. Have the authors made all data underlying the findings in their manuscript fully available?

Reviewer #1: Yes

Reviewer #2: No

4. Is the manuscript presented in an intelligible fashion and written in standard English?

Reviewer #1: Yes

Reviewer #2: Yes

5. Review Comments to the Author

Reviewer #1: Evaluations

1. In general, how do you rate the degree to which the paper is easy to follow and its logical flow? Good

2. Do the title and abstract cover the main aspects of the work? Yes

3. If relevant, are the methods are clear? No

Study design, ethical issues, and data analysis are mixed. Data were collected through social media (Facebook). Most of the time, Facebook users are uses by hiding or changing their correct name. So, what you say on quality of data, the real individuals these participated in your study? And so on… Generally, might be individuals are use fake account for Facebook. Clarify

4. If relevant, did the authors, make the underlying data available to the readers? Yes

5. Are the tables clear and legible? No

All tables’ titles are not self-explanatory. Rewrite all the tables as they are self-explanatory and easily understandable.

6. Does the paper raise any ethical concerns? Yes

7. Are the figure clear and legible? No

Title of the figure, rewrite under the figure and as scientifically sounded.

8. Most literatures review has no references. Please check it through the documents.

9. In table 6, you were listed gender (other), ethnicity (other). What is other?

10. Acknowledgements are too long.

11. I have not seen the importance of separating references of table 2 from the rests.

12. In table 6, what does it means by “First Episode Psychosis and Other psychosis disorder?” it is not clear for the readers especially by DSM-5, it has no meaning?

13. In table 6, you summarized that the age range =(18-74), mean = (41.51), and SD= (12.35). Did you check for normal distribution of the age? Think more about it. I think it is not appropriate. Same comment for Černis Felt Sense of Anomaly in the table 6.

Reviewer #2: The manuscript introduces a new theoretical framework for understanding dissociation, namely Felt Sense of Anomaly (FSA). They then use this concept to develop a questionnaire for measuring FSA. The call the scale, the ČEFSA, the Černis Felt Sense of Anomaly Scale. They don’t explain until well into the article (page 14) that the concept of FSA comes from their qualitative study of dissociative experiences in psychosis (Černis, Freeman & Ehlers, 2020). My biggest criticism is that the authors don’t make clear from the outset that the purpose of yet another dissociation questionnaire is for their study of dissociative symptoms in psychotic disorders. They misrepresent it as an instrument that will perform “housecleaning” in the field of dissociation.

They start with a literature review of all the studies that did factor analyses of dissociation instruments. The review would have been more useful if they had also described the samples tested (i.e., diagnosis and level of trauma exposure) and their average Dissociative Experiences Scale (DES) scores. (This is the same rationale as to why the authors included the DES and Posttraumatic Symptom Disorder Checklist (PCL) in their instrument development study.) This would help to explain the differing results in the studies. I also have a fundamental problem with the methodology of reviewing the previous instruments by looking at the different factors comprising dissociation in order to select items for a new instrument. By doing this, they are discounting the studies that reported dissociation as a single factor. With this methodology, they are not at all looking at the sub-factors such as dissociative amnesia (DA) and self-states that go into the single factor. (For example, Dell outlined 12 sub-factors in the Multidimensional Inventory of Dissociation; Dell, 2006). Furthermore, it is known that pathological dissociation, as found in the dissociative disorders (DDs) is a taxon (Waller, Putnam & Carlson, 1996), a different phenomenon than “normal” dissociation, which is a dimensional phenomenon. Pathological dissociation is a unitary phenomenon and therefore largely excluded from these authors’ new instrument. Understanding the population sampled in the studies reviewed would give clues as to whether the samples are experiencing pathological or non-pathological dissociation and whether they are traumatized or not (since trauma correlates with dissociation).

When it came to developing the framework for the new questionnaire, they used their conclusions from their qualitative study of dissociative experiences in psychotic disorders (Černis, Freeman & Ehlers, 2020) namely that dissociation is commonly experienced as a felt sense of anomaly. It seems they only used the literature review to substantiate their theory, rather than as a basis for their new instrument. Then they developed a completely new framework to understand FSA dissociation: different “domains” (physical sensation. perception, mental content or selfhood) can be affected by a type of “anomaly” (unfamiliarity, unreality, automaticity, lack of control, or sense of detachment or absence). This produced an 8 x 5 matrix and the authors “generated” two items per cell as well as global items. Using repeated exploratory factor analysis on a sample recruited from the internet, they reduced the number of items to 35, which included 7 factors.

It is puzzling why the authors then piloted the new instrument in a population of psychotic disorders. They justify it by saying that dissociation is a “transdiagnostic” symptom. I would say that dysphoria and anxiety are also transdiagnostic symptoms, but one wouldn’t pilot depression or anxiety questionnaires in a population of psychotic disorders. One should pilot a dissociation questionnaire in subjects with dissociative disorders. I suspect that the agenda is that they are building on their qualitative study of dissociative experiences in psychosis (Černis, Freeman & Ehlers, 2020). I understand the importance of understanding dissociative symptoms in the development and maintenance of psychosis. However, I feel that claims that this new instrument performs “housecleaning” in the field of dissociation at large, and reduces some of the confusion that surrounds the construct of dissociation, are over-exaggerated. They did mention in three places in the article that they were delineating a discrete subset of dissociative experiences. Therefore, this is not a comprehensive measure of dissociation. Hence adding a “novel measure of dissociative experiences” that is not comprehensive to the literature, is adding to the confusion, not reducing it.

I appreciate the discussion of why dissociative amnesia and compartmentalized parts were not included in their instrument (see excerpt below). However, I think their dismissal of DA as controversial by quoting one article and not summarizing the copious literature supporting “frank” DA as an important dissociative symptom is grossly biased. A more balanced review of the literature is warranted as well as an honest and detailed explanation of why “frank” DA is beyond the scope of their questionnaire. The point is that “frank” DA and dissociated parts of self are the diagnostic criteria for Dissociative Identity Disorder and the hidden agenda of this study is to describe and quantify dissociative symptoms in psychosis. Of course their qualitative study of dissociation (Černis, Freeman & Ehlers, 2020) did not uncover “frank” DA, therefore it was not included in this instrument.

“Whilst detachment or unfamiliarity of memory does fall within the framework of FSA-type dissociation, frank dissociative amnesia (such that another ‘part’ of the personality retains a memory that is entirely inaccessible by another ‘part’) is not. However, despite it being regarded as a cardinal symptom of dissociation by many, the topic of dissociative amnesia remains controversial [6]. Therefore, we emphasise that FSA-type dissociation does not preclude the possibility of dissociative amnesia, but includes itself only those experiences where memory is experienced with a subjective sense of strangeness. Experiences where the person completely loses awareness of their current surroundings or responds to content in memory as if it represented the present are also not captured by the questionnaire.” (p.34-35).

I request that the authors make the following changes:

• Make clear from the outset that the purpose of this dissociation questionnaire is for the study of dissociative symptoms in psychotic disorders.

• Remove all references to their instrument performing “housecleaning” in the field of dissociation.

• Remove references to the ČEFSA reducing the confusion that surrounds the construct of dissociation

• Add to the literature review chart a description of the samples tested: diagnosis, level of trauma exposure, and their average DES scores.

• A more balanced review of the literature around “frank” Dissociative Amnesia and an honest and detailed explanation of why compartmentalization-type DA is beyond the scope of their questionnaire.

Černis E, Freeman D, Ehlers A. Describing the indescribable: A qualitative 559 study of dissociative

560 experiences in psychosis. PLoS One. 2020;15:e0229091

Dell, P. (2006). The Multidimensional Inventory of dissociation (MID): a comprehensive measure of pathological dissociation. Journal of Trauma & Dissociation, 7(2), 77-106.

Waller, N. G., Putman, F. W., & Carlson, E. B. (1996). Types of dissociation and dissociative types: a taxometric analysis of dissociative experiences. Psychological Methods, 1, 300-321.

6. PLOS authors have the option to publish the peer review history of their article (what does this mean?). If published, this will include your full peer review and any attached files.

Reviewer #1: **Yes: **Zakir Abdu

Reviewer #2: No

---

## [Author Response · Author response to Decision Letter 0]

1 Dec 2020

Dear Dr Sar,

Re: A new perspective and assessment measure for common dissociative experiences: ‘Felt Sense of Anomaly’.

We are very grateful to the two reviewers for their kind comments and thoughtful feedback on the above manuscript.

Please find detailed below our responses to these, as requested in your email dated 9th November 2020. Please note that we have also updated Part 2 (clinical study) of this manuscript following the receipt of additional data. This data has not changed the results or conclusions of the manuscript.

I hope that following the changes made to the manuscript - as outlined below - this manuscript now meets the standard required for publication in Behavioural and Cognitive Psychotherapy.

Yours sincerely,

Dr Emma Černis

Wellcome Trust Clinical Doctoral Fellow & Clinical Psychologist

 

Journal requirements

We thank the editor for highlighting these requirements and have amended our manuscript accordingly.

No minors were included in this study. All participants were aged 18 or over.

3. Please describe in your methods section how capacity to consent was determined for the participants in this study.

This has now been added to the methods sections for both parts of the manuscript.

4. In your Data Availability statement, you have not specified where the minimal data set underlying the results described in your manuscript can be found. PLOS defines a study's minimal data set as the underlying data used to reach the conclusions drawn in the manuscript and any additional data required to replicate the reported study findings in their entirety. All PLOS journals require that the minimal data set be made fully available. For more information about our data policy, please see http://journals.plos.org/plosone/s/data-availability .

Upon re-submitting your revised manuscript, please upload your study’s minimal underlying data set as either Supporting Information files or to a stable, public repository and include the relevant URLs, DOIs, or accession numbers within your revised cover letter. For a list of acceptable repositories, please see http://journals.plos.org/plosone/s/data-availability#loc-recommended-repositories . Any potentially identifying patient information must be fully anonymized.

Important: If there are ethical or legal restrictions to sharing your data publicly, please explain these restrictions in detail. Please see our guidelines for more information on what we consider unacceptable restrictions to publicly sharing data: http://journals.plos.org/plosone/s/data-availability#loc-unacceptable-data-access-restrictions . Note that it is not acceptable for the authors to be the sole named individuals responsible for ensuring data access.

Data cannot be shared publicly because of the terms and conditions contained within the ethics permissions granted for this study from the Central Research Ethics Committee of the University of Oxford, the NHS Research Ethics Committee, and Health Research Authority, and consented to by participants. Surveys were confidential to enable freedom of expression by participants, and participants consented into the study without being consulted as to the sharing of anonymised data. Therefore, only descriptive statistics, which qualify as the minimal data set, are included in the paper.

 

Reviewer: 1

1. In general, how do you rate the degree to which the paper is easy to follow and its logical flow? Good

2. Do the title and abstract cover the main aspects of the work? Yes

3. If relevant, are the methods are clear? No

Study design, ethical issues, and data analysis are mixed. Data were collected through social media (Facebook). Most of the time, Facebook users are uses by hiding or changing their correct name. So, what you say on quality of data, the real individuals these participated in your study? And so on… Generally, might be individuals are use fake account for Facebook. Clarify

We thank the reviewer for highlighting this and have now separated the design and ethics for each study more clearly. The reviewer raises a valid point about the quality of Facebook data. The limitations of using Facebook for research is discussed in the Discussion section of the paper, where we have now added further text to address the points raised here.

4. If relevant, did the authors, make the underlying data available to the readers? Yes

5. Are the tables clear and legible? No

All tables’ titles are not self-explanatory. Rewrite all the tables as they are self-explanatory and easily understandable.

We have now reviewed all table titles, and amended for clarity.

6. Does the paper raise any ethical concerns? Yes

We are sorry that the reviewer believes our paper raises ethical concerns. We hope that our responses to other reviewer comments and subsequent changes to the manuscript serve to reassure the reviewer that this study was carried out ethically and with full consideration of participants’ experience of the research.

7. Are the figure clear and legible? No

Title of the figure, rewrite under the figure and as scientifically sounded.

We have now reviewed all figure titles, and amended for clarity.

8. Most literatures review has no references. Please check it through the documents.

We have checked the manuscript thoroughly, and are now satisfied that all references are in place.

9. In table 6, you were listed gender (other), ethnicity (other). What is other?

We thank the reviewer for this question. 

With regards to gender, we are aware from participant feedback on previous research run by our group that respondents may not identify with binary gender descriptors and required another option(s) to reflect this. Since there are many descriptors incorporated within the gender non-binary lexicon, and the proportion of respondents selecting these was likely to be very small, we felt it was appropriate to gather all gender non-binary options under ‘other’. This is also in-keeping with Stonewall (stonewall.org.uk) guidelines, which recommend a ‘prefer to self-describe’ option.

With regards to ethnicity, we were aware that even a broad range of options for this question would not capture all respondents’ ethic identities, and therefore included the option of ‘other’ and a free-text field. Examples of ‘other’ given by respondents were various mixed-heritage or multiracial identities, as well as Jewish, Latinx, European and Arab.

10. Acknowledgements are too long.

We agree with the reviewer that this is a long acknowledgements section. However, we feel that it is important to recognise the hard work of all the NHS trusts who were integral to the success of this study. We hope that the generous page allowance of PLoS One will allow us to retain this important section in full.

11. I have not seen the importance of separating references of table 2 from the rests.

We thank the reviewer for this thorough consideration. These references are separated from those for the main body of the manuscript due to the large number of references in Table 2 (n=77). We have noted other authors using this method of separating systematic review references into an appendix, and therefore considered it appropriate to do the same in this manuscript. This has the benefits of shortening the length of the manuscript from the perspective of formatting, and makes the references in the main body of the manuscript easier for the reader to find in the references section.

12. In table 6, what does it means by “First Episode Psychosis and Other psychosis disorder?” it is not clear for the readers especially by DSM-5, it has no meaning?

We thank the reviewer for highlighting this. These are clinical terms commonly used within the UK mental healthcare system that relate to diagnostic categories within the ICD 10. ‘First Episode Psychosis’ (FEP) describes a psychosis presentation that is new (first ever in the person’s medical history) and requires the patient to begin antipsychotic medication. Usually, this is coded as F29, ‘Unspecified psychosis not due to a substance or known physiological condition’. The option ‘unspecified’ is used for FEP as the presentation may be too recent to determine its chronicity i.e. whether criteria are met for Brief Psychotic Disorder (F23), or whether criteria are met for code Schizophrenia (F20). The equivalent of FEP in the USA is ‘Early Psychosis’. This category is well-established within the psychosis clinical research field. ‘Other psychosis disorder’ relates to code F28 defined as ‘other psychotic disorder not due to a substance or known physiological condition’ and includes chronic hallucinatory psychosis.

13. In table 6, you summarized that the age range =(18-74), mean = (41.51), and SD= (12.35). Did you check for normal distribution of the age? Think more about it. I think it is not appropriate. Same comment for Černis Felt Sense of Anomaly in the table 6.

Reporting the ranges, means and standard deviations is good practice in research. Furthermore, the size of this sample (n over 1000) constitutes adequate response spread and statistical power that the analysis carried out here is valid. In support of this statement, we highlight that both the Bartlett’s test of Sphericity and Kaiser-Meyer-Olkin tests of sampling adequacy supported the use of factor analysis with this data. Nevertheless, age was checked for Normal distribution and outliers. Age followed a Normal distribution with no significant level of skew (skewness coefficient = 0.084). The ČEFSA data was also checked, and a slight skew was found (skewness coefficient = 0.63). This is not uncommon in symptom scales and we refer the reader back to the aforementioned sample size and pre-analysis checks.

 

Reviewer: 2

I request that the authors make the following changes:

• Make clear from the outset that the purpose of this dissociation questionnaire is for the study of dissociative symptoms in psychotic disorders.

We thank the reviewer for this valuable reflection on our manuscript. It is not our intention that the ČEFSA measure is specific to the study of dissociative symptoms in psychotic disorders. In response to the reviewer’s interpretation of our work, we have now amended the text to more clearly explain the relationship of the qualitative study to the decision-making process regarding the FSA matrix and item generation. We have also amended the text to reflect that whilst the new scale is tested here with a psychosis group, dissociation has well-established links to other presentations, and the scale will require further testing within these groups in future.

We hope that the development of the scale items with consideration of a broad range of input (the systematic review), and using non-clinical (general population) data satisfies the reviewer that, whilst valid for use in a psychosis group, our scale is not specific to this context.

• Remove all references to their instrument performing “housecleaning” in the field of dissociation.

Both references to this have now been removed.

• Remove references to the ČEFSA reducing the confusion that surrounds the construct of dissociation

We thank the reviewer for this helpful feedback and have now removed these references. We aimed to portray to the reader that the approach taken in this study is only one perspective, and that we can only know if this conceptualisation is helpful over time and with further testing. We hope that the manuscript now more closely reflects this stance.

• Add to the literature review chart a description of the samples tested: diagnosis, level of trauma exposure, and their average DES scores.

We thank the reviewer for this suggestion, and have now added a column in Table 2 entitled ‘sample characteristics’, which reports the number, population, PTSD diagnosis rate, and DES mean and standard deviation where possible.

• A more balanced review of the literature around “frank” Dissociative Amnesia and an honest and detailed explanation of why compartmentalization-type DA is beyond the scope of their questionnaire.

We thank the reviewer for the opportunity to expand on this important point. We have now added a more detailed discussion of the limitations of the concept of FSA-dissociation and the ČEFSA scale with respect to dissociative amnesia, and to compartmentalisation symptoms more generally. This additional text is intended to clarify to the reader that it is not our intention to discount dissociative amnesia as an important construct, but to highlight that - as a new perspective requiring further testing and exploration - it is not yet clear how the conceptualisation presented in this manuscript may relate. We hope that this additional text now more closely reflects this intention.

---

## [Decision Letter · Decision Letter 1]

11 Jan 2021

PONE-D-20-20041R1

A new perspective and assessment measure for common dissociative experiences: ‘Felt Sense of Anomaly’

PLOS ONE

Dear Dr. Cernis,

Thank you for submitting your manuscript to PLOS ONE. After careful consideration, we feel that it has merit but does not fully meet PLOS ONE’s publication criteria as it currently stands. Therefore, we invite you to submit a revised version of the manuscript that addresses the points raised during the review process.

We look forward to receiving your revised manuscript.

Kind regards,

Vedat Sar, M.D.

Academic Editor

PLOS ONE

Reviewers' comments:

Reviewer's Responses to Questions

**Comments to the Author**

1. If the authors have adequately addressed your comments raised in a previous round of review and you feel that this manuscript is now acceptable for publication, you may indicate that here to bypass the “Comments to the Author” section, enter your conflict of interest statement in the “Confidential to Editor” section, and submit your "Accept" recommendation.

Reviewer #1: (No Response)

Reviewer #2: (No Response)

2. Is the manuscript technically sound, and do the data support the conclusions?

Reviewer #1: Partly

Reviewer #2: Yes

3. Has the statistical analysis been performed appropriately and rigorously? 

Reviewer #1: Yes

Reviewer #2: I Don't Know

4. Have the authors made all data underlying the findings in their manuscript fully available?

Reviewer #1: No

Reviewer #2: Yes

5. Is the manuscript presented in an intelligible fashion and written in standard English?

Reviewer #1: No

Reviewer #2: Yes

6. Review Comments to the Author

Reviewer #1: Title

1. Does the title give clear idea about the article? Yes

Abstract

2. Does the abstract concisely describe the content and scope of the project and identifies the project’s objective, its methodology and its findings, conclusions, or intended results? No

Under background, add something about dissociation more or its impact.

Introduction

3. Does the introduction give clear idea about the article? YES

Under introduction clarify “ […]”

Methods

4. Did method part is clear? No

Well, if specified who (author/s) were extracted data.

What do you say about data quality assessment? Because, Participants were recruited via social media.

Is all original paper report standard error? If no, what is your action? If yes, please specify.

Results

5. Are results clear and appropriate with title? Yes

6. Revise the references as per the journal guideline

7. The paper needs an English language copy editing from the beginning to the end. Please focus on it.

8. Generally, the paper is interesting. The methodology part needs revision.

Reviewer #2: The language of the abstract is very convoluted. After rereading the “Definition and Framework Development section” several times, I think the abstract should be rewritten.

They talk about a “common denominator” and a construct” and don’t say from the outset that it was a Felt Sense of Anomaly (FSA) that they were looking to delineate from the beginning, based on their previous research. This convoluted language implies that FSA was discovered in the data that forms the basis of this paper.

Also, discussion of “common denominator” contradicts the concept of “discrete subset”. “Common denominator” suggests that they aim to find the one factor that underlies all of dissociative phenomena. The use of the term “common denominator” should be accompanied by the qualification that it is a common denominator of a subset of dissociative experience.

I suggest the following revision of ideas to the background and methods sections of the abstract (using phrases from the Definition and Framework Development section):

Background

The term ‘dissociation’ has long been argued to lack conceptual clarity and may describe several distinct phenomena. Based on our previous qualitative research into the lived experience of dissociation in people with psychosis diagnoses, we found that dissociation is commonly experienced as a subjective FSA. We therefore aimed to conceptualise and empirically delineate thisa discrete subset of dissociative experiences based on a common denominator and develop a corresponding assessment measure.

Methods

First, a systematic review of existing measures was carried out to identify a common denominator themes amongst dissociative experiences, including FSA phenomena. Second, Iitem generation for the the new measure was based on this construct.literature review and our previous qualitative study. Third, Mmeasure development was achieved using exploratory (EFA) and confirmatory (CFA) factor analysis of 8861 responses to an online self-report survey. Fourth, Tthe resulting measure was then validated via CFA with data from 1031 NHS patients with psychosis diagnoses.

It would also assist with the transparency of their goals if they discussed their qualitative study “Describing the indescribable” in the Introduction section. Most authors describe the evolution of their ideas that led to the current paper by reviewing their own publications in the Introduction.

Thank you for redoing Table 2 with a column that describes the sample more completely. Now that I realize that your goal in the review was more to find FSA phenomena in the literature rather than find one common denominator for all of dissociative phenomena, I think describing PTSD in the samples reviewed is unnecessary and can be deleted.

I appreciate the discussion of the relationship of Dissociative Amnesia to FSA.

7. PLOS authors have the option to publish the peer review history of their article (what does this mean?). If published, this will include your full peer review and any attached files.

Reviewer #1: No

Reviewer #2: No

---

## [Author Response · Author response to Decision Letter 1]

19 Jan 2021

Please see "Response to Reviewers" letter attached.

---

## [Editor Report · Decision Letter 2]

1 Feb 2021

A new perspective and assessment measure for common dissociative experiences: ‘Felt Sense of Anomaly’

PONE-D-20-20041R2

Dear Dr. Cernis,

We’re pleased to inform you that your manuscript has been judged scientifically suitable for publication and will be formally accepted for publication once it meets all outstanding technical requirements.

Kind regards,

Vedat Sar, M.D.

Academic Editor

PLOS ONE
---

## [Editor Report · Acceptance letter]

5 Feb 2021

PONE-D-20-20041R2 

A new perspective and assessment measure for common dissociative experiences: ‘Felt Sense of Anomaly’ 

Dear Dr. Černis:

I'm pleased to inform you that your manuscript has been deemed suitable for publication in PLOS ONE. Congratulations! Your manuscript is now with our production department. 

Kind regards, 

on behalf of

Dr. Vedat Sar 

Academic Editor

PLOS ONE